# Study on Extra Virgin Olive Oil: Quality Evaluation by Anti-Radical Activity, Color Analysis, and Polyphenolic HPLC-DAD Analysis

**DOI:** 10.3390/foods10081808

**Published:** 2021-08-05

**Authors:** Francesco Cairone, Stefania Petralito, Luigi Scipione, Stefania Cesa

**Affiliations:** Department of Drug Chemistry and Technology, “La Sapienza” University of Rome, Piazzale Aldo Moro, 5, 00185 Rome, Italy; francesco.cairone@uniroma1.it (F.C.); stefania.petralito@uniroma1.it (S.P.); luigi.scipione@uniroma1.it (L.S.)

**Keywords:** extra virgin olive oils, polyphenols, carotenoids, HPLC-DAD analysis, spectrophotometric analysis, reflectance colorimetry, quality evaluation

## Abstract

This study aimed to evaluate the quality of oils available on the Italian market and purchased directly from the mill or in the supermarket and labelled as extra virgin olive oils (EVOOs). As one of the most relevant foods of the Mediterranean diet and recognized as a functional food if regularly consumed, the quality of EVOO needs to be continuously monitored. Different analytical protocols were applied. The spectrophotometric parameters used to classify the extra virgin olive oils—a CIEL*a*b*color analysis and the quali-quantitative analysis of bioactive molecules by HPLC-DAD detection and the anti-radical activity, by the DPPH method, were evaluated and compared among the samples. This study confirmed a very high variation in terms of quality, both in oils purchased directly from mills throughout Italy, but also in oils labeled as “100% of Italian origin”. Due to the high variability reconfirmed in the monitored samples, it is necessary to carry out a capillary control, not limited only to the parameters indexed by law. A useful complementary method could be represented by reflectance colorimetric analysis.

## 1. Introduction

The prevention of cardiovascular diseases, hypertension, diabetes, inflammation, oxidative stress, gut microbiota alteration, and liver disease exerted by the traditional Mediterranean diet, enriched with further integration of extra virgin olive oils (EVOOs), nuts, and pistachios, is well-known and recognized by the whole scientific community [1,2,3,4].

The multiple protective effects exerted by EVOO, the main fat source of the Mediterranean diet and commonly recognized as a functional food, are widely reported and generally attributed to its high content of minor components. The phenyl alcohols tyrosol and hydroxytyrosol (*p*-(hydroxyphenyl)ethanol and 3,4-(dihydroxyphenyl)ethanol), the secoiridoid oleuropein, flavonols, and lignans, all belonging to the polyphenolic fraction, as well as the balance among monounsaturated, saturated, and polyunsaturated fatty acids and the antioxidant potential exerted by tocopherols and carotenoids, can contribute to the health benefits associated with EVOO consumption [5,6,7,8,9].

The CEE Regulation n. 2568/91 [10], completed and revised by the EU Regulation 1604/2019 [11], defines all the parameters required to classify an extra virgin olive oil. Besides belonging to this class, extra virgin olive oils could present many different characters in terms of color, bioactive content, antioxidant properties, and retail price. Moreover, the EU Regulation 432/2012 [12] defines the health claims for olive oil polyphenols indicating that “olive oil polyphenols contribute to the protection of blood lipids from oxidative stress”, but also specifies that “the claim may be used only for olive oil which contains at least 5 mg of hydroxytyrosol and its derivatives per 20 g of olive oil” and that “the beneficial effect is obtained with a daily intake of 20 g of olive oil”.

Color analysis, and more specifically CIEL*a*b* analysis, of extra virgin olive oil was performed by several authors, in combination with other analytical methods to evaluate the designation of protected designation origin (PDO) products. Becerra-Herrera et al. [13] evaluated the color character and the phenolic profile of nine Spanish extra virgin olive oils belonging to the new cv. “Serrana de Espadan”, concluding that tyrosol, hydroxytyrosol, and their derivatives, together with the lignin pinoresinol and the total polyphenol content (TPC) were the most important parameters to take into account for the quality evaluation. Kosma et al. [14] studied a large number of Greek oils, belonging to less-known Greek cultivars. Combining data, relative to quality parameters, fatty acid composition, and color analyses, they evidenced significant and statistically relevant differences among the various considered cultivars.

The phenolic content, as well as the tocopherol, carotenoid, and secoiridoid content of PDO and monovarietal oils of Italian origin, were detected and evaluated in different studies, from which it was possible to extrapolate median values for all these characterizing parameters and obtain a preliminary overview of the quality exerted by extra virgin olive oils, generally recognized as the golden standard for the olive production [15,16]. From the reported data, it was possible to observe an average content of phenols covering the range 250–610 µg/g oil, among which decarboxymethyl elenolic acid linked to hydroxytyrosol (3,4-DHPEA-EDA) was the most represented molecule. Remarkable quantities of α-tocopherol (160–235 µg/g oil) and carotenoids (β-carotene and lutein, 4–10 µg/g oil) were also reported [15,16].

As is well known, an important relationship exists between agricultural practices, procedural technological steps, and high quality of final products, as explained in different papers that show the correlation between applied systems and the expressed phytocomplex contents. In the review by Lanza et al. [17], a series of intensive procedures related to EVO oil production and their effects on the secoiridoid and phenol content were presented. In particular, a significant loss of natural antioxidants was reported after pasteurization and during storage since the heat treatment impacts primarily on secoiridoids. Antonini et al. [18] conducted a study on 28 extra virgin olive oils, comparing two different oil extraction methods. One involved the use of a two-way decanter and the other a conventional three-way decanter. The 14 oils produced with the two-way decanter revealed the presence of higher contents of secoridoids, lignans, and tyrosols.

On the basis of these findings, the current work aimed to evaluate the quality of oils available on the Italian market and purchased directly from the mill or in supermarkets, and labeled as extra virgin olive oils (EVOOs), by applying different analytical protocols. The spectrophotometric parameters of the UV region, required to classify extra virgin olive oils [11], the color measure by reflectance analysis, the quali-quantitative analysis of the most important bioactive molecules, and the overall anti-radical capacity of the supplied oil samples were measured. The resulting data were compared to obtain a picture of the current situation in Italy, confirming a very high variation in terms of quality.

Moreover, it was evaluated if, and to which extent, the color analysis, based on a simple, quick, and economic procedure and directly executable in the field by unskilled personnel, could return a quality evaluation of such a precious food commodity.

## 2. Materials and Methods

### 2.1. Materials

A total of 54 samples of extra virgin oil were analyzed. Of these, 28 were purchased directly by mills located in nine Italian regions Abruzzo (A1), Basilicata (B1), Campania (C1–C3), Lazio (L1–L13), Molise (M1), Puglia (P1–P5), Sicily (S1), Tuscany, (T1 and T2), and Umbria (U1), and were designed as the MILL series; 13 oils (Ar, Au, Ca, Cc, Ce, Cp, Ct, Cv, Ds, Fb, Ma, Pa, and Sa) which came from large retailers and were labelled as “100% Italian origin”, were designed as the ITA series and 13 oils (Aa, Ac, Ai, Be, Cf, Cm, Co, Cr, Dc, Dn, Gi, Lr, and Sp), which came from large retailers and were labelled as “olive oils by the European Union”, were designed as EUR series (Appendix A). All samples were stored in the dark at room temperature until the analyses were performed.

Bidistilled water, methanol, acetone, acetic acid, acetonitrile, *n*-hexane, and reference compounds (gallic acid, hydroxytyrosol, ferulic acid, oleuropein, and 1,1-diphenyl-2-picrylhydrazyl (DPPH)) for HPLC were purchased from Merck life Science s.r.l. (Milan, Italy).

### 2.2. Spectrophotometric Analysis

#### 2.2.1. K_232_, K_270_ and ΔK Parameter Analysis

Fifty milligrams of oil were solubilized in 5 mL of *n*-hexane and analyzed by UV/VIS Lambda 25 spectrophotometer (Perkin Elmer, Waltham, MA, USA) in the ultraviolet range. Parameters K_232_, K_270_, and ΔK, quality indicators according to Regulation (EEC) No 2568/91 [10], were determined at wavelengths of 232 nm and 270 nm, respectively. ΔK corresponds to the value given by the following operation: K_268_ − (K_262_ + K_274_/2).

#### 2.2.2. Carotenoid and Chlorophyll Analysis

One hundred milligrams of sample were solubilized with 5 mL of *n*-hexane and analyzed by UV/VIS Lambda 25 spectrophotometer (Perkin Elmer Waltham, MA, USA), in the visible range (400–700 nm). Absorbance values, at 470 and 670 nm, were recorded according to Minguez-Mosquera et al. [19].

#### 2.2.3. DPPH (2,2-diphenyl-1-picryl-hydrazyl) Assay

According to Cioffi et al. [20], with slight modifications, 7.0 mg of DPPH were solubilized in 100 mL of propan-2-ol. Then, 2 mL of this solution were added to 1 mL of propan-2-ol, stored in the darkness, and monitored by UV/VIS Lambda 25 spectrophotometer (Perkin Elmer Waltham, MA, USA), at the wavelength of 515 nm, until the absorbance value was stable. The value of the maximum absorption expressed by the radical was read after 20′. Then, 37.5 mg of oil were weighed, solubilized in 1 mL of propan-2-ol, and added to the cuvette along with 2 mL of the same DPPH solution. The absorbance at 515 nm was determined, following the same conditions described above and the reduction of DPPH absorbance after 20′ was evaluated. Finally, a calibration curve was constructed to quantify the antioxidant activity by adding 1 mL of gallic acid (from 0.23 to 8.28 µg/mL), at different concentrations, to 2 mL of the DPPH solution following the previous described conditions. A calibration curve was constructed (y = −0.276 ln(x) + 0.6143) and the antioxidant capacity exerted by the tested oil samples was expressed as gallic acid equivalents.

### 2.3. Colorimetric Analysis

The extra virgin olive oil samples were analyzed for their color character, with an X-Rite SP-62 colorimeter (X-Rite Europe GmbH, Regensdorf, Switzerland), set with a D65 illuminant and a 10° observer angle, as previously described [21]. Each experiment was performed four times and the results are expressed as the mean value ± standard deviation (SD). Cylindrical coordinates C*_ab_ and h_ab_ were calculated from a* and b* as in [22].

### 2.4. Solid-Phase Extraction and HPLC-DAD Analysis of Polyphenols

Extra virgin olive oil samples were subjected to solid-phase extraction using a Discovery^®^ DSC-18 SPE Tube column (Merck Life Science, S.r.l., Milan, Italy), according to Mateos et al. [23], with substantial modifications. The column was previously conditioned with *n*-hexane. About 4.0 g of oil was dissolved in 10 mL of *n*-hexane and loaded into the column. The column was washed with 6 mL of *n*-hexane and 6 mL of acetonitrile which were discarded, and 6 mL of methanol for the extraction of the polyphenolic content. The obtained methanol fractions were concentrated under reduced pressure, at a controlled temperature of 40 °C, with a rotary evaporator, weighed, and stored at 4 °C until HPLC-DAD analyses were performed.

The obtained extracts were solubilized in methanol. The chromatographic analyses were performed on a Luna Phenyl-Hexyl column (150 × 4.6 mm, 5 µm) using an HPLC (Perkin Elmer Waltham, MA, USA) apparatus consisting of a Series 200 LC pump, a Series 200 DAD, and a Series 200 autosampler, including a Totalchrom Perkin Elmer software for the data acquisition. The mobile phase consisted of a mixture of 1:1 methanol/acetonitrile (solvent A) and water acidified with 5% acetic acid (solvent B) in solvent gradient, from 5% to 70% of solvent A in 55′, at a flow rate of 1 mL/min. Analyses were conducted at 280 nm for the identification of phenolic acids and secoiridoids and at 360 nm for the identification of flavonoids. Calibration curves were constructed for hydroxytyrosol (R^2^ = 0.9983), ferulic acid (R^2^ = 0.9974), oleuropein (R^2^ = 0.9979), and quercetin-3-D-galactoside (R^2^ = 0.9999) which were quantified.

### 2.5. Liquid–Liquid Extraction and HPLC-DAD Analysis of Carotenoids

Four selected extra virgin olive oil samples Au (ITA), Cf (EUR), A1, and L13 (MILL, Abruzzo and Lazio) were subjected to liquid–liquid extraction according to Minguez-Mosquera et al. [24], with modifications. First, 2.0 g of oil was dissolved in 20 mL of *n*-hexane, loaded in a separating funnel, and extracted with 20 mL of acetonitrile three times. The acetonitrile fractions were concentrated under reduced pressure, at a controlled temperature of 40 °C, with a rotary evaporator, weighed and stored at 4 °C until HPLC-DAD analyses were performed.

Organic extracts enriched in carotenoids were weighed and solubilized in methanol. The analyses were performed according to Patsilinakos et al. [25]. Chromatography was performed on Luna C18 column (150 × 4.60 mm, 3 µm) at 450 nm using an HPLC Perkin Elmer apparatus, as described above. The mobile phase consisted of a mixture of 95:5 methanol/water acidified by 5% acetic acid (solvent A) and acetone (solvent B) at 90% of solvent A, in isocratic mode, at a flow rate of 1 mL/min.

### 2.6. Statistical Analysis

All the performed analyses were reproduced in quadruplicate. The data obtained were statistically validated using the system (ANOVA) with attached standard deviation. Principal component analysis (PCA) was performed using XLSTAT 2020 version (Addinsoft Inc, New York, NY, USA) software.

## 3. Results and Discussion

### 3.1. Spectrophotometric Analyses

The 54 samples of extra virgin olive oils (sample legend in Appendix A), purchased from the Italian market were classified into three clusters on the basis of their provenance (by mill, cluster MILL) and label (by supermarket, “100% of Italian origin”, cluster ITA; or “blend of oils coming from the European Union”, cluster EUR). The 28 samples classified as “MILL” came from nine different Italian regions and showed visibly different colors and grades of turbidity, whereas the 26 samples from supermarket, 13 labelled as “of Italian origin” and 13 labelled as “of European Union origin” appeared similar for color and clarity. To these classes, different prizes were associated. The oils purchased from Italian mills were, on average, as expensive as the ITA cluster, whereas the EUR cluster were 20–25% (or more) less expensive.

Spectrophotometric analyses were performed on samples at two different concentrations, aiming to evaluate the absorption value in the UV region (K_232_, K_270_, and ΔK) and the visible zone of the spectrum for carotenoids (about 470 nm) and chlorophylls (about 670 nm). All data, reported in Appendix A, show that some samples of each cluster were out of the accepted regulatory range, but nevertheless labeled as EVOO. Some samples with spectrophotometric values of K_232_ and K_270_ significantly out of range were identified. L11, P4, and P5 showed a K_232_ > 3.17, whereas Pa, Ma, and T2 showed a K_232_ > 2.78 (vs. a Regulation Limit of 2.5). Among these, the samples L11, P4, P5, T2, and Pa also had K_270_ > 0.26 (vs. a Regulation Limit of 0.22). The EUR sample Dc, despite a low K_232_ value, showed an exceptionally high K_270_ value of 1.46 and a ΔK of 0.032 (Regulation Limit of 0.01). The K_270_ of the other seven samples ranged between 0.25 and 0.32 and the ΔK of the other three samples ranged between 0.031 and 0.053.

Many papers are reported in literature in which spectrophotometric analysis in the UV region was used to evaluate the quality of extra virgin olive oil, but these generally refer to the control of a small number of samples and/or to selected cultivars, or to the evaluation of the shelf-life [26,27]. In other studies, the extinction coefficient was correlated with other quality parameters such as the metal content [28], the anti-radical capacity, and the polyphenol content during ripening [29], or with the peroxide value [30].

With respect to the carotenoid and chlorophyll absorption regions, MILL samples showed very different absorption values at 470 nm, ranging between 0.002 and 0.115. In addition, the ITA and EUR samples ranged, in a narrower region, between 0.015 and 0.040 and between 0.013 and 0.037, respectively. Similarly, absorption values at 670 nm ranged between 0.001 and 0.099 in MILL, in a narrower range of 0.009 and 0.046 in ITA, and in an even narrower range of 0.010 and 0.032 in EUR samples. Usually, higher absorption values at 670 nm correspond to a higher carotenoid contents read at 470 nm.

The chlorophyll absorption of <0.015 corresponded to the carotenoid absorption of, on average, 0.018 (22 samples); the chlorophyll absorption of <0.030 corresponded to the carotenoid absorption of, on average, 0.025 (23 samples); and the last group had a carotenoid mean of 0.046 (9 samples). In two MILL samples (C1 and B1), an exceptionally low chlorophyll content was revealed (<0.002) with respect to a mean minimum value of 0.004. In addition, in L13 and L2 an exceptionally high value (>0.070) with respect to a mean maximum value of 0.020 was shown. In conclusion, in relation to the pigment content, the MILL samples, although presenting a mean pigment content (0.047 ± 0.041) comparable to that of ITA samples (0.049 ± 0.013) and higher with respect to the EUR series (0.020 ± 0.006), were exceptionally dispersed, as furtherly confirmed by the colorimetric parameters.

Spectrophotometric analyses were also performed to evaluate the anti-radical potential of the oil samples by using DPPH assay. Opportune DPPH solutions were monitored for 20′, in the darkness at controlled temperature, until stable absorbance values were read at 515 nm. Known oil amounts were then added and their scavenging activity against this radical was monitored in the same operative conditions. Finally, a calibration curve with gallic acid was constructed and results were expressed as µg equivalents of gallic acid/g oil (Appendix A). Results are reported in Figure 1, panel A, as mean values of the three different clusters. The ITA cluster presented the highest DPPH value, but this was due to five samples which showed values higher than 200, as well as another six samples presenting very low values, ranging between 13 and 26 µg equivalents of gallic acid/g oil, thus denoting a very high sample dispersion. For this reason, results were also reported (Figure 2, Panel B) selecting four different groups on the basis of the different DPPH value. In each group, the following clusters were represented: low DPPH value, 13 samples, among which there were 5 MILL, 6 ITA, and 2 EUR; mean DPPH value, 18 samples, among which there were 9 MILL, one ITA, and eight EUR; high DPPH value, 16 samples, among which there were 13 MILL, 1 ITA, and 2 EUR; very high DPPH value, 7 samples, among which there were1 MILL, 5 ITA, and 1 EUR. This demonstrated that the anti-radical capacity of the analyzed samples was not correlated with the cluster to which they belonged.

### 3.2. Color Analysis

The oils were analyzed as such, not filtered or clarified, by colorimetric CIEL*a*b* analysis, and the data are reported in Appendix A. Data showed a high dispersion for the L* (luminance) values of cluster MILL (between 36 and 56). On the contrary, a similar narrow region was shown by the other two classes (46–55 for cluster ITA and 44–53 for cluster EUR). The character a* of color, green for negative and red for positive values, was very near to 0 value, denoting a grey region and the almost total absence of this kind of pigments, even if some samples seemed yellow-greenish, for the presence of chlorophylls. On the contrary, the positive, and often quite relevant b* value denoted a yellow character which assumed highly different values among samples, ranging between 6 and 52. Oil from mills, and those of 100% Italian origin (ITA) had higher variabilities with values ranging between 6 and 50 and 9 and 52, respectively, as well as b* values for the EUR oils ranging between 31 and 52, in a definitely narrower region of the color space.

A valuable method to deepen the character of color differences among samples was represented by the principal components analysis (PCA). This was carried out on all the data collected. The values were processed through XLSTAT 2020 software, building a biplot with 98.13% of correlation related to CIEL*a*b parameters as shown in Appendix A. The principal components analysis (PCA) confirmed the high dispersion of MILL samples with respect to the little and almost overlapping regions of ITA and EUR samples. PCA analysis allowed us to identify samples that largely deviated from the reference clusters. B1, C3, L10, Cf, and Fb showed L* > 54 with respect to a mean value of 49.31 ± 4.60; Cf and Fb also showed b* > 50 with respect to a mean value of 37.84 ± 11.27; and B1, C3, and L10 showed a h_ab_ > 89 with respect to a mean value of 85.66 ± 2.02. In conclusion, the reflectance curves were plotted as cluster mean values, after discarding these five samples and the seven samples previously evidenced as they were out of range for the spectrophotometric parameters (L11, P4, P5, T2, Ma, Pa, and Dc).

Reflectance curves (Appendix A) denoted only small differences, not supported by statistical significance between classes, with an intermediate profile of the ITA samples, with respect to the inferior curve of MILL comprehending slightly darker samples, and the superior curve of EUR involving slightly lighter samples. The differences among the three identified classes were not significant as the standard deviations were particularly high, especially for MILL samples, but also for ITA samples. On the contrary, EUR samples were highly defined in a restricted color region, which probably accounts for a slightly higher impact during the manufacturing [31].

The MILL samples were furtherly divided into subclasses on the basis of chlorophyll and carotenoid content after discarding the out-of-range samples (C1, L2, L13). As expected (Figure 2), the cluster with “high pigment content” (HPC MILL) was characterized by a lower reflectance curve (Abs_470_ + Abs_670_ between 0.05 and 0.21), corresponding to a darker color as well as a “low pigment content” (LPC MILL) (Abs_470_ + Abs_670_ between 0.01 and 0.04) showing a reflectance curve overlapping with those from ITA samples having slightly higher pigment content (Abs_470_ + Abs_670_ between 0.03 to 0.07). Higher carotenoid contents were generally aligned with higher chlorophyll content. Kosma et al. [14] reported L* values between 66 and 72, b* values between 70 and 98, and more negative a* values between −5 and −9. Conversely, Piscopo et al. [32] and Sicari et al. [33], investigating the effect of storage on EVO oils, found CIEL*a*b parameters in line with our results. Despite the fact that, through the study of color, it could be possible to differentiate oils by cultivar and geographical origin [13,14], there are no studies in which a correlation of the expressed color with the oil quality has been attempted.

We attempted a correlation to understand if the content of pigments (carotenoids and chlorophylls), and thus the shown color, could be associated with a higher or lower quality of oil, in terms of contribution to health. Four clusters were identified in relation to their pigment content and the relative reflectance curves were evaluated as mean values (Figure 2). No one evident correlation was shown among these clusters, with the expressed value of bioactive molecules, or with the anti-radical capacity. The presence of a higher carotenoid content, accompanied to a higher content of chlorophylls, manifested as a lower reflectance curve, is not correlated to a higher antioxidant protective role. Chlorophylls, generally denoting a less impactive production technology, could even play an adverse role, in terms of protection from the autoxidation process.

The mill samples C1 and B1 showed an exceptionally low carotenoid and chlorophyll content, which maybe denotes an impacting process. The highest values were found in five MILL samples, which came from the Lazio region, are maybe ascribable to the geographic origin.

### 3.3. Polyphenolic and Carotenoid HPLC-DAD Analyses

Two different methods for the selective extraction of the minor components represented by the polyphenolic and carotenoid content were adopted. The selective extraction of the polyphenolics (phenolic acids, secoiridoids, and flavonols) was performed according to Mateos et al. 2001 [23], by using solid-phase extraction on reverse phase RP18, with some modification, allowing us to obtain extraction yields ranging from 0.5 to 5% in the majority (80%) of the samples analyzed.

The obtained extracts, analyzed by HPLC-DAD analyses, gave a characteristic profile (Appendix A), by which some molecules, such as oleuropein, quercetin, and kaempferol derivatives were identified by comparison with literature data [19,34]. Tyrosol, hydroxytyrosol, oleuropein, quercetin, and ferulic acid also could be quantified on the basis of the calibration curves. The recorded chromatograms showed the typical polyphenolic pattern of olive oil [35], in which hydroxytyrosol (1), ferulic acid (3), and oleuropein (4) with absorption at 280 nm, and quercetin-3-galactoside (6) with absorption at 360 nm were identified by external standard as the main peaks. Comparing the chromatograms with those in the literature, it was possible to identify other peaks—tyrosol (2), oleuropein derivative (5), quercetin derivative (7), and kaempferol derivatives (8) [23]. Peaks 7 and 8 were quantified and expressed as quercetin-3 D-galactoside equivalents (Appendix A).

Data obtained by the analyses of all the treated samples are reported in Appendix A. The content was expressed as µg/g oil. Hydroxytyrosol ranged between 1.27 and 35.6, tyrosol between 1.45 and 39.1, ferulic acid between 0.28 e 4.09, oleuropein and derivatives between 0.82 and 70.40, and flavonols between 0.03 and 1.33. These data are only partially confirmed in literature, in which hydroxytyrosol was found in a narrower range between 20.6 and 34.6, tyrosol between 0.54 and 7.93, ferulic acid between 0.13 and 0.68, oleuropein and derivatives between 19.90 and 140.00, and quercetin and derivatives between 0.27 and 17.40 [15,16,19,34].

The content of the polyphenols in the three different clusters, generally recognized as a quality marker, is reported in Figure 3. A decreased incidence of cardiovascular diseases, obesity, and cancer is particularly correlated with the recognized health properties of tyrosol, quercetin, and oleuropein derivatives and with their antioxidant preventive effects [17,18,36].

By this comparison, it was shown that all the three classes had similar mean values of hydroxytyrosol, tyrosol, and quercetin. The ITA samples also showed much higher levels of secoridoids and ferulic acid. Quercetin was more represented both in the ITA and in the EUR samples with respect to the MILL samples. Overall, a higher content of tyrosols, secoridoids, and flavonols was furtherly associated with higher anti-radical activity, with some exceptions for the samples Fb (ITA), Cm (EUR), and L6 and L10 (MILL), where the high anti-radical capacity could depend on the possible presence of other active molecules.

Based on what is reported in literature [16,23,37,38], the typical carotenoid EVOO pattern mainly consists of β-carotene and lutein. To confirm this profile, some of our samples were selected. These were submitted to a purification step in order to obtain a selective extraction of the carotenoid components. According to the method optimized by Minguez-Mosquera et al. [23], four samples (A1, L13, Au, and Cf), selected from the three different clusters on the basis of the different absorbances expressed at 415 nm, were submitted to a liquid–liquid extraction, whose performance was based on the ability of acetonitrile to selectively extract the slightly less lipophilic carotenoid content by the oily phase dissolved in *n*-hexane The so-obtained purified extracts were analyzed by HPLC-DAD following the procedure described by Patsilinakos et al. [25] and the relative chromatograms are reported in Appendix A. The adopted extraction method allowed us to obtain only the less lipophilic carotenoid moiety, represented by xanthophylls, while the residue carotenes could be remained in the *n*-hexane phase. Despite the highly different absolute carotenoid content, the four analyzed samples showed the same xanthophyll profile, in which more than 90% of peak areas was represented by lutein, confirming the overall data shown by literature.

### 3.4. Statistical Analysis

Data evaluation and PCA analysis of the monitored variables allowed us to build a quality rank, by which it was possible determine the five best samples (in the red triangle in Figure 4), the ITA samples, Ds, Cp, Ca, and Ce and the MILL L8, selected for the highest sum value (DPPH + HPLC data considered as sum) between 218 and 338 and the five worst samples (in the blue triangle), the MILL samples P2, M1, L11, U1, and L10, selected for the lowest sum value, ranging between 9 and 18 (very high values of polyphenol content and DPPH scavenging activity were also found for ITA sample Sa, but this was discarded for its out-of-range ΔK value). In the three circles (violet, green, and red), the regions corresponding to increasing values of quality are evidenced and it is possible to show that the three clusters are represented in each region. The three discarded samples Fb, L6, and Cm denoted an exceptionally high ratio between DPPH value and HPLC content, in part related to the very low HPLC content, which could denote a possible presence of active molecules coming from compounds of a different nature and maybe index of some adulteration. The attempt to correlate the parameter for each color to the other detected values did not afford the expected results, but it was possible partially correlate the lowest oil quality with the highest h_ab_ values (Appendix A). Finally, selecting all the outsider samples with respect to the others (in agreement with the regulatory limits and presenting mean values for all the evaluated parameters) and plotting the corresponding reflectance curves, low reflectance values at 670 nm were finally associated with best oil quality (Appendix A).

## 4. Conclusions

Several of the analyzed samples, all purchased in the Italian market or directly from the producing mills, as well as those labelled as extra virgin olive oils, were out of range with respect to the spectrophotometric parameters indexed by law. Significant differences among samples were evidenced with respect to the carotenoid and chlorophyll content. The pigment content, other than related to the olive origin, cultivar, geographic, and agronomic parameters, could also be deeply influenced by the applied treatments. This was particularly evident in the MILL samples, where a lack of standardization was reflected in a higher dispersion.

A very high variability was also found in terms of bioactive molecules and expressed anti-radical capacity; these two parameters did not seem correlated. On the contrary, most samples with the lower biomolecule content (<20 µg/g) were also out of range for spectrophotometric parameters. An exceptionally high ratio between DPPH value and HPLC content could denote a possible presence of active molecules coming from different matrices. The highest h_ab_ values correlated only in part with some of the worst samples. In addition, lower values of reflectance at 670 nm were partially correlated with samples of the best quality. On the whole, even if not predictive, the color analysis allowed us to identify a narrower range of parameters associated with the best quality samples and deserves to be further investigated.

Samples labelled as “oils of European origin” were more standardized with respect to the all analyzed parameters and characterized, as a cluster, by the lowest standard deviations.

Belonging to the three identified classes (purchased by mill, of Italian origin, or of European origin) did not justify the significant differences found in terms of prizes awarded, which were not sustained by quality parameters. These are, in fact, exceptionally dispersed in the samples from mills and “of Italian origin” and more homogeneous in the samples “of European origin” which showed intermediate quality parameters.

Reflectance colorimetry, which is quick, simple, economic and does not require experts, could be considered a valid method for a preliminary screening of quality, allowing samples with too high L* and b* values to be discarded, but many experiments should still be performed with the aim of identifying a reflectance curve to use as fingerprint of quality EVOOs.

## Figures and Tables

**Figure 1 foods-10-01808-f001:**
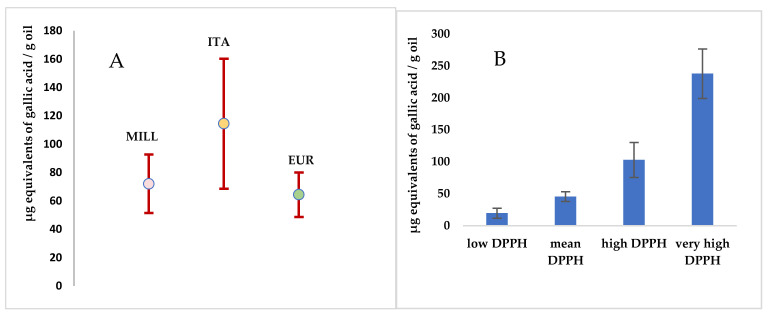
Comparison of anti-radical capacity reported as mean values of the three selected clusters (Panel **A**) and as mean values of four different classes selected by DPPH content (Panel **B**).

**Figure 2 foods-10-01808-f002:**
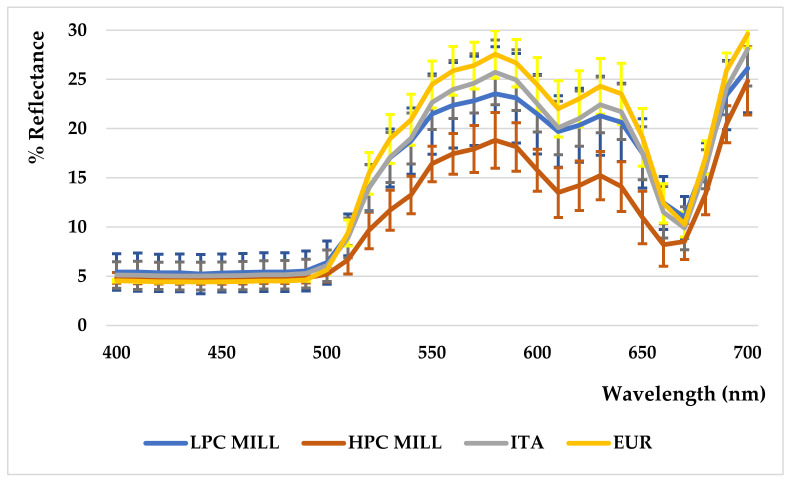
Reflectance curves of selected samples on the basis of carotenoid and chlorophyll content.

**Figure 3 foods-10-01808-f003:**
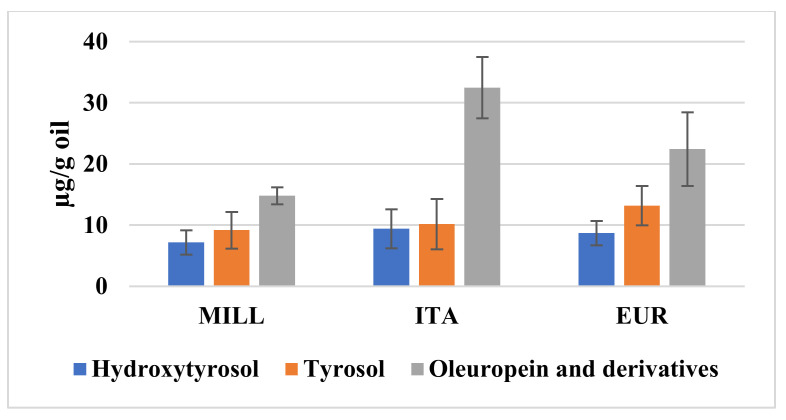
Average values of the main polyphenolic components in the three main clusters (MILL, ITA, and EUR).

**Figure 4 foods-10-01808-f004:**
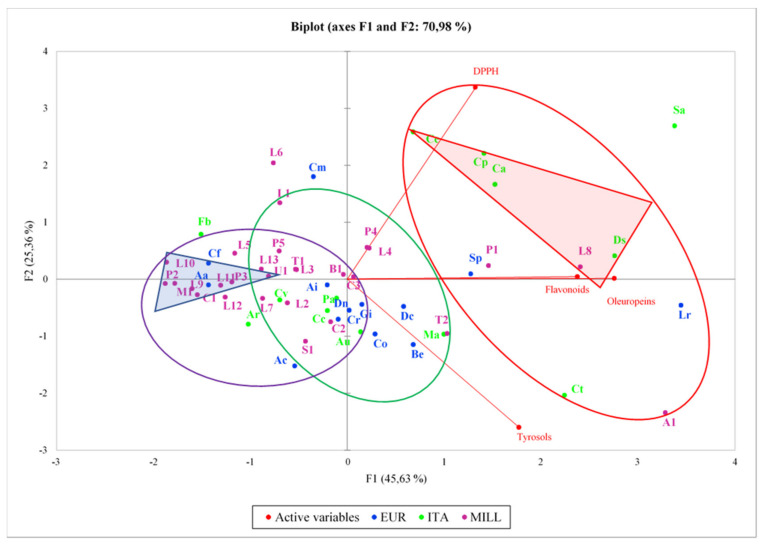
Principal component analysis (PCA) of the all analyzed samples related to monitored variables.

## Data Availability

The datasets generated for this study are available on request to the corresponding author.

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
