# Peer review of "Study on Extra Virgin Olive Oil: Quality Evaluation by Anti-Radical Activity, Color Analysis, and Polyphenolic HPLC-DAD Analysis"

_foods, 2021, doi:10.3390/foods10081808_

Round 1

Reviewer 1 Report

Thank you for addressing all the suggestions and for improving your manuscript. 

Reviewer 2 Report

I have no additional comments.

Best regards

Reviewer 3 Report

The article entitled: "Study on extra virgin olive oil: quality evaluation by anti-radical activity, color analysis and polyphenolic HPLC-DAD analysis" has been reviewed again and corrections made correctly. Although I have not received notification of the responses from the authors, corrections have been made to the article.

Have the authors considered doing other analyses such as acidity level and peroxide content to evaluate fatty acids and sensory attributes to verify organoleptic evaluation (e.g., fruitiness, bitterness and pungency)?

Reviewer 4 Report

The authors have addressed my concerns.

This manuscript is a resubmission of an earlier submission. The following is a list of the peer review reports and author responses from that submission.

Round 1

Reviewer 1 Report

In my opinion it is a very interesting study. Extra virgin olive oil is a popular product consumed in Europe, mostly in Mediterranean countries and it brings many health benefits.

It is really valuable that the authors decided to study a lot of different olive oils (54 samples) as well from the mill as well from the market, however a better characterization of the materials is missing in my opinion.

Additionally, it is quite confusing that f.ex. in HPLC-DAD analysis the authors selected only 4 samples out of 54. I think such a low number is not very representative.

 In the Results, part of the manuscript a deeper discussion (comparison with other studies) of the results should be improved.

A small editing mistakes:

Line 70, remove a dot after reported “reported.[15, 16].”

Replace in the whole paper “ml” with “mL” (capital letter when mention Liter)

Line 332: “showed very higher” replace with much higher

Reviewer 2 Report

The manuscript needs some improvements.

  • Figure 1. Please present the antioxidant activity in the same way as the polyphenols content in Fig. 3.
  • Please correct the description of the y axis in figure 3.

Reviewer 3 Report

There are some changes that should be made.  I list them below:

  1. Introduction

I have no objections to the Introduction, it is concise and well referenced.

  1. Materials and Methods

Line 62: What does it mean “DPO” ? Maybe it means “PDO”

Line 95: Table S1 should be described in the order in which it appears in the table itself, i.e. the first group of oils should be the 28 oils from the mills. Then the group of oils from Italy and finally the European oils.

Lines 120, 125, 129: the word "Two" should be a number (2)

Line 128: The term "I mL" is not clear, for a better understanding I suggest indicating the range of concentrations of the standard gallic acid.

Line 160: I do not understand why liquid-liquid extraction for carotenoid analysis is done only for 4 oils.

  1. Results and Discussion

Line 178: Discussion is missing.

Discussion of Spectrophotometric analysis should be supported by the literature not only by regulation limits.

Line 223: the result of the gallic acid curve should be expressed in μg equivalents of gallic acid /g oli as shown in table S2.

Line 225- 228: In the DPPH sample count there are a total of 53 samples, however, as described in line 95 there are 54 samples. Thus there is one sample that is not being counted or it is missing.

Figure S1: the sample labeled B1 appears as B in the graph.

Figure 3: the ordinate axis is written in Italian.

Line 334: “Eur” must be written in capital letters.

  1. Conclusions

Conclusions section is correct

Reviewer 4 Report

Manuscript foods-1274407 presents an evaluation of olive oils from Italian origin, European origin, and oils direct from Italian mills using UV spectrophotometry to assess carotenoids and chlorophylls, reflectance colorimetric analysis, DPPH radical-scavenging ability, and HPLC-DAD analysis for polyphenolics.  Based on these analyses, the olive oils of Italian origin showed high variability and many were outside the legal parameters for extra virgin olive oil.  Olive oils of European origin, on the other hand, presented lower standard deviations in the measurements.  There does not seem to be a correlation between secondary metabolite concentrations and radical-scavenging activity.  This report has presented complementary methods to assess quality of olive oil.

It is not clear what the message is for Figure 1.  Wouldn't it be better to have the MILL, ITA, and EUR DPPH values (along with SD bars) included in this graph?

The manuscript should be proof-read by an English-speaking technical editor.
